# Analytical Evaluation of Wet and Dry Mechanochemical Syntheses of Calcium-Deficient Hydroxyapatite Containing Zinc Using X-ray Diffractometry and Near-Infrared Spectroscopy

**DOI:** 10.3390/pharmaceutics14102105

**Published:** 2022-10-01

**Authors:** Makoto Otsuka, Hanae Saito, Tetsuo Sasaki

**Affiliations:** 1Research Institute of Electronics, Shizuoka University, 3-5-1 Johoku, Naka-ku, Hamamatsu 432-8011, Japan; 2Faculty of Pharmacy, Musashino University, 1-1-20 Shinmachi, Nishi-Tokyo, Tokyo 202-8585, Japan; 3Graduate School of Medical Photonics, Shizuoka University, 3-5-1 Johoku, Naka-ku, Hamamatsu 432-8011, Japan

**Keywords:** zinc-containing calcium phosphate compounds, controlled zinc release, mechanochemical synthesis, wet and dry grinding, near-infrared spectroscopy, transformation to hydroxyapatite

## Abstract

Calcium-deficient zinc-containing calcium phosphate (ZnAP), which has sustained zinc release properties that are effective for treating osteoporosis, can be efficiently synthesized as a biomaterial through wet grinding. To elucidate the physicochemical mechanism of these mechanochemical syntheses, ground products were obtained from the starting material powder (S-CP), consisting of calcium hydrogen phosphate dihydrate (CHPD), calcium oxide (CaO), and zinc oxide (ZnO), by wet and dry grinding for 0–3 h in a centrifugal ball mill. The ground S-CP products were analyzed using powder X-ray diffraction (XRD) and near-infrared spectroscopy (NIRS); the crystal transformations and molecular interactions of the ground products were kinetically analyzed. The XRD and second-derivative NIRS results indicate that the S-CP is primarily transformed into ZnAP via amorphous solid formation in wet grinding, and the reaction follows a consecutive reaction model. In contrast, in dry grinding, the ground product of CHPD and CaO is transformed into an amorphous solid following an equilibrium reaction model; however, ZnO is predominantly not transformed and remains crystalline.

## 1. Introduction

Hydroxyapatite (HAP) [1,2] and its related compounds [3,4] have been developed as bioactive bone materials with excellent biocompatibility that can be directly connected to bones and are available on the market as various implant preparations. However, indications for these implants in treating the elderly and patients with osteoporosis are currently insufficient. The focus of these recent biomaterial studies has been to develop more biologically active materials; therefore, the characteristics of bone remodeling by osteoclasts and osteoblasts in transplanted bones must be comprehensively understood [5]. The specific drug delivery system (DDS) [6,7], which consists of applying a self-setting apatite cement, works as an artificial bone in hard tissues with affinity for natural bone and controls bone metabolism by slowly releasing bone growth factors from within the device. This artificial bone material with intelligent DDS function evidently demonstrates a significant therapeutic effect on bone mineral density in rats with osteoporosis [8,9,10,11,12,13].

Sixteen minerals—namely, Ca, P, K, S, Cl, Na, Mg, Zn, Cr, Co, Se, Fe, Cu, Mn, Mo, and I—are essential for supporting life [14]. These minerals need to be supplied from external sources to maintain vital activities, such as physiological functions and metabolism in the human body. As these essential minerals are highly bioactive, symptoms of deficiency are induced by limited consumption and overdose by excessive consumption. In particular, Zn is an essential trace element and a cofactor in more than 200 enzymes; Zn is present in almost every cell in the body and is used as a metabolic factor [15].

Zn deficiency is definitively diagnosed when the “serum Zn level” is less than 60 μg/dL with accompanying “clinical symptoms” [14]. Zn deficiency is associated with dysgeusia, anemia, dermatitis, aphthous ulcers, alopecia, intractable pressure ulcers, loss of appetite, stunted growth, gonad dysfunction, infertility, and susceptibility to infection [16,17,18,19,20,21,22,23,24,25,26,27,28,29,30,31,32,33,34,35]. In bone tissue, Zn inhibits osteoclast differentiation, promotes osteoblast activity, and significantly affects the formation of hard tissue [36]. Therefore, Zn administration has been reported as a clinical treatment for osteoporosis and Zn deficiency in the elderly and some patients [37,38].

However, as excessive Zn dosage can result in serious side effects in terms of cytotoxicity [14,24], consistently ingesting small amounts over extended durations is pharmacologically important. Therefore, the allowable range of Zn concentration in the living body for obtaining the optimum therapeutic response is very narrow, and using a DDS that appropriately controls the Zn concentration is necessary in order to increase the bone density and improve the therapeutic effect [37,38].

An ion substitution method for essential trace metal ions in calcium phosphate crystals was proposed for the first time as a unique DDS through basic science [39]. The controlled slow release of Zn incorporated into the tricalcium phosphate (Ca_3_(PO_4_)_2_) crystal structure properly induced bone formation in animal experiments [39]. Subsequently, several Zn-containing calcium phosphate complexes have been synthesized, and their controlled Zn release requires long durations while avoiding the side effects of Zn [40,41,42]. The physicochemical properties and Zn release behaviors of these series of Zn-containing calcium phosphate compounds and their applications in osteoporosis and wound treatment have been demonstrated in animal experiments [43,44].

In general, wet chemical reactions which are consumed a large amount of energy, and emit highly toxic organic solvents, waste liquids and exhaust gas including carbon dioxide are became a hindrance factor for the world environmental [45]. The United Nations has recommended industrial use of green chemistry to reduce negative impacts on the future global environment as a Sustainable Development Goal [46]. Chemical reactions based on mechanochemistry could be recommended as ideal green chemistries without high energy consumption per unit weight and producing waste solvents. The solid-state reactants in mechanochemical synthesis [47,48,49,50] are ground and absorbing the minimum energy required to initiate the reaction in a ball mill and efficiently proceeding with the reaction to the product without any toxic waste solvents and gases.

Studies on the mechanochemical synthesis of calcium phosphates by wet grinding have revealed that Ca^2+^ replaces Sr^2+^ in this process [51,52,53]. In the previous report [54], Zn-doped calcium phosphate (ZnAP) was prepared via a simple wet mechanochemical synthesis using centrifugal agate ball milling. Crystalline properties of ZnAP were sufficiently characterized by qualitative methods, such as powder X-ray analysis, scanning electron microscopy with energy-dispersive X-ray spectroscopy, the nitrogen gas adsorption and in vitro Zn release test, and the results clearly indicated that the formation of Ca-deficient HAP containing Zn during grinding [54]. Biomedical activities of the ZnAP were optimal ranges for bone regeneration medicine by evaluating various pharmaceutical properties. An obtained injectable suspension formulation using ZnAP can be applied to any parts of the body. At the injection site, the nanoparticles with bioaffinity are phagocytosed by cells such as macrophages, and the particles dissolved and release Zn [54], so, sustained Zn release rate could be freely controlled. The result indicated that the injectable ZnAP suspension formulation may be useful practical dosage form for the slow release of Zn. However, the detailed molecular formation mechanism in the wet and dry mechanochemical synthesis of therapeutically useful ZnAP has not been clarified yet.

In this study, therefore, the effects of the wet or dry grinding environment on the synthetic mechanism of the calcium phosphate complex containing Zn from the mixed powder of the raw materials and their molecular interactions were investigated using kinetic methods based on powder X-ray diffraction (XRD) and near-infrared spectroscopy (NIRS).

## 2. Materials and Methods

### 2.1. Materials

Calcium oxide (CaO; Lot No. LAK5373), calcium hydrogen phosphate dihydrate (CHPD; CaHPO_4_·2H_2_O; Lot No. SDR3931), and zinc oxide (ZnO; Lot No. LAF2459) were purchased from FUJIFILM Wako Pure Chemical Corporation (Tokyo, Japan).

### 2.2. Synthesis of Zinc Calcium Phosphate Powders

A physically mixed powdered sample (CaO (1.169 g):CHPD (8.104 g):ZnO (0.727 g) = 7:20:3, molar ratio) was used as the starting powder material (S-CP). The wet method involved grinding S-CP (10 g) with 10 mL of purified water in an agate planetary ball mill (Pulverisette 6, Fritsch Japan Co., Yokohama, Japan) containing 10 agate balls (diameter = 20 mm), whereas the dry method involved grinding the powder without water. Different sample preparations were obtained by grinding in the ball mill at a centrifugal speed of 200 rpm at room temperature (approximately 20–25 °C) for 0, 0.5, 1.0, 1.5, 2.0, 2.5, and 3.0 h. All ground preparations were heated in a furnace (TMF5T; THOMAS, Tokyo, Japan) at 70 °C for 5 h.

### 2.3. Powder XRD Analysis

All the test samples were crystallographically characterized using XRD (RINT-Ultima-III, Rigaku Co., Tokyo, Japan; Cu Kα radiation, 40 kV, 40 mA, scan speed = 4°/min). The XRD area was obtained by integrating the recorded XRD profiles for *2θ* = 5–45°. The XRD %area (XRD%) of the ground S-CP product was evaluated based on the total area of the XRD-time profile between 2θ = 5 and 45° using the XRD computer software JADE (Materials Data Inc., Livermore, CA, USA); the XRD% of the initial S-CP was assumed to be 100% crystalline.

### 2.4. Dissolution Test

The in vitro dissolved amount of Zn in the sample powders was measured in phosphate-buffered saline (PBS; pH 6.2). The dissolution test was performed using the paddle method at 100 rpm in 200 mL of PBS at 37.0 ± 0.5 °C [54]: Each powder sample (20 mg) was examined using a dissolution tester (NTR-6100, Toyama Sangyo Co. Ltd., Osaka, Japan), and the sample solution (1 mL) was collected after 10 min. The Zn concentration was quantified using a commercially available Zn assay kit (5-Br-PAPS, Fuji Film Chemical Industry Co., Tokyo, Japan) with an ultraviolet–visible absorption spectrometer (UV/VIS; UV2550, Shimadzu Co., Kyoto, Japan) at 560 nm. The in vitro data represent the average of three measurements.

### 2.5. NIRS Measurement

NIRS was measured using an NIRS instrument (MPA, Bruker Optics Japan Co., Yokohama, Japan) with a diffused fiber-optic probe; the measurement conditions were as follows: range of measurement = 7000-3700 cm^−1^; resolution = 16 cm^−1^; scanning time = 48 scans. NIR spectra were pretreated using the second-derivative function [55,56].

## 3. Results

### 3.1. Changes in XRD Profiles of Ground S-CP Products during Wet and Dry Grinding

Figure 1 shows the XRD patterns of the S-CP containing CHPD, CaO, and ZnO (20:7:3) under wet and dry grinding conditions. In wet grinding, the diffraction peaks of all raw materials significantly decrease with increasing grinding time. In the XRD pattern of the product obtained by grinding for more than 2 h, broad diffraction peaks at 26.0 and 32.0° ascribed to HAP are observed, as reported previously [54]. In contrast, in dry grinding, the diffraction peaks attributed to CHPD and CaO significantly decrease after approximately 1 h of grinding; however, the intensity of the original ZnO diffraction peak remains unchanged. Moreover, the intensities of the HAP peaks do not increase, even after grinding for 3 h.

Figure 2 displays the effects of wet and dry grinding on the XRD% of the ground S-CP products. In wet grinding, the XRD% of the ground product decreases to approximately 40% after 30 min of grinding and approximately 20% after 1.5 h of grinding. In contrast, in dry grinding, the XRD% of the ground product is approximately 60%, which decreases to 30% after 3 h of grinding.

Figure 3 depicts the change in the XRD peak height intensities of individual raw materials in the S-CP during wet or dry grinding. In wet grinding, the peak intensity of CHPD significantly decreases to approximately 10% or less within 1 h; however, the peak intensities of ZnO and CaO gradually decrease with increasing grinding time. The peak intensity of HAP increases for up to 1 h of grinding and then presents a constant value at approximately 300 cps. In contrast, in dry grinding, the peak intensity of CHPD rapidly decreases to less than 10% in 30 min. However, the peak intensities of ZnO and CaO exhibit a slight decrease and that of HAP minimally increases, despite grinding for 3 h.

Figure 4 shows the relationship between the XRD peak height intensities of individual raw materials in the ground S-CP products obtained by wet or dry grinding and their XRD%. In wet grinding, the peak intensities of CHPD and ZnO decrease and that of HAP increases with decreasing XRD%. In contrast, in dry grinding, the peak intensity of CHPD sharply decreases with decreasing XRD% and that of ZnO slightly decreases; the peak intensity of HAP does not increase.

### 3.2. Changes in NIR Spectra of the Ground S-CP Products during Wet and Dry Grinding

Figure 5 shows the NIR spectra of the starting materials and the resultant S-CP products obtained at various grinding times. The NIR spectra of the original starting materials are shown in Figure 5a, and changes in the spectra of the S-CP consisting of CHPD, CaO, and ZnO (20:7:3) during wet grinding are depicted in Figure 5b. In particular, the absorption peaks at approximately 7000, 5000, and 4000 cm^−1^ (Figure 5c–e, respectively) are significantly altered during grinding. Therefore, the peak changes in specific bands are analyzed using second-derivative spectral treatment to clarify the molecular interactions in the ground products during wet and dry grinding (Figure 6).

Figure 7 shows the changes in peak intensities at 7000, 5000, and 4000 cm^−1^ in the second-derivative spectra of the ground S-CP products during wet and dry grinding, based on Figure 6. In wet grinding, the peak intensities at 5180 and 5110 cm^−1^ increase with increasing grinding time. The intensities at 7085 and 3876 cm^−1^ increase after 30 min and remain constant with additional grinding time. In dry grinding, the peak intensities at 7085, 5180, 5110, and 3876 cm^−1^ increase with increasing grinding time. Moreover, the intensity change rate at 7085 and 5180 cm^−1^ for the ground products obtained by wet grinding is considerably faster than that by dry grinding. However, the intensity change rates at 5110 cm^−1^ are approximately equal for wet and dry grinding.

Figure 8 shows the relationships between peak intensities at specific bands in the second-derivative NIR spectra of the ground S-CP products during wet or dry grinding and their XRD%. In wet grinding, the peak intensity at 5180 cm^−1^ slowly increases with decreasing XRD%, whereas in dry grinding, the intensity increases at an accelerated rate with decreasing XRD%. The peak intensity at 7085 cm^−1^ sharply increases with decreasing XRD% during wet grinding, whereas it slowly increases during dry grinding. The peak intensities at 5110 and 3876 cm^−1^ slightly increase with decreasing XRD% during wet and dry grinding.

## 4. Discussion

### 4.1. Effect of Environmental Conditions on Mechanochemical Transformation Pathways of the Ground S-CP Products

ZnAP with sustained Zn release could be easily synthesized from S-CP powder consisting of CHPD, CaO, and ZnO via the wet mechanochemical method in this study, similar to the findings reported in our previous study [54]; however, it was not obtained under dry conditions. Therefore, in this study, the chemical reaction pathways of individual raw powder materials in the S-CP powder during wet and dry grinding and their molecular interaction behaviors during ZnAP synthesis were determined using NIRS.

Abdeslam et al. [57] kinetically investigated the mechanism of crystal transformation from a physical mixture of CHPD and CaO into HAP by wet and dry grinding. They reported that CHPD was transformed into Ca-deficient HAP [Ca_10−x_(HPO_4_)_x_(PO_4_)_6−x_(OH)_2−x_·*n*H_2_O] via an amorphous solid [Ca_3_(PO_4_)_2_·*x*H_2_O] as an intermediate. In addition, Miyaji et al. [58] reported that Zn could replace Ca in HAP by up to 15 mol% through precipitation, and Zn-substituted HAP was obtained as [Ca_10x_(Zn)_10(1−x)_(PO_4_)_6_(OH)_2_].

In this study, the crystal transformation mechanism of ZnAP during grinding is considered based on the solid-state mechanochemistry of HAP [57] as follows: In wet grinding, CaO, which is highly water-soluble, is immediately transformed into Ca(OH)_2_ because of its instability in a wet environment (Figure 3), whereas the relatively stable ZnO slowly transforms into Zn(OH)_2_ in an alkaline wet mass. This is possibly because the solubility of ZnO in water is approximately 700 times lower than that of CaO [59,60]. In the XRD profiles (Figure 1 and Figure 2), the crystal diffraction peaks attributed to CHPD in the S-CP sharply decrease within approximately 1 h of grinding, and approximately 80% or more of the solid is transformed into an amorphous state. In addition, the peak intensity of ZnO slowly decreases to approximately 30% during wet grinding. Therefore, because an unstable amorphous solid reacts with Ca^2+^ in the wet grinding environment, the solid is gradually transformed into Ca-deficient HAP with low crystallinity [61,62]. Zn^2+^ is possibly incorporated into the HAP structure when the amorphous solid is transformed into Ca-deficient HAP; thus, the product contains Zn with low crystallinity.

Moreover, based on the relationship between the XRD peak intensity of each component and the XRD% of the ground products (Figure 4), Zn is primarily incorporated into the Ca-deficient HAP crystal during the recrystallization of the amorphous solid after CHPD and Ca(OH)_2_ are amorphized by grinding.

The ZnAP complex formation mechanism during wet grinding can be summarized as follows:

0.7CaO + 0.3ZnO + H_2_O → 0.7 Ca^2+^ + 0.3 Zn^2+^ + 2OH^−^

2 CaHPO_4_·2H_2_O + 0.7 Ca^2+^ + 0.3 Zn^2+^ + 2HO^−^

→  1/3Ca_8.1_Zn_0.9_(PO_4_)_6_ + 6 H_2_O

→  1/3Ca_9.1−x_Zn_0.9_(HPO_4_)_x_(PO_4_)_6−x_(OH)_2−x_ + *n*H_2_O    *x* = 1

→  1/3Ca_8.1_Zn_0.9_(HPO_4_)(PO_4_)_5_(OH) + *n*H_2_O

However, in dry grinding, the diffraction peak intensity of CHPD significantly decreases within 30 min owing to amorphization with dehydration by mechanochemical energy. Approximately 80% or more of CHPD is transformed into an amorphous solid and free water (Figure 2, Figure 3 and Figure 4) [63]. Owing to its stability in a dry environment, the amorphous solid does not transform into HAP. The diffraction peak intensity of ZnO is not significantly reduced by dry grinding (Figure 3), and it is predominantly not transformed into hydrates owing to a little water in the solid. The amorphous solid reacts with CaO to produce a Ca-deficient amorphous solid [Ca_3−x_(HPO_4_)_x_(PO_4_)_2−x_·*n*H_2_O]. Therefore, in dry grinding, CHPD is rapidly transformed into an amorphous solid; however, ZnO primarily remains in the solid because it is not recrystallized into HAP.

The ZnAP complex formation mechanism during dry grinding can be summarized as follows:

2 CaHPO_4_·2H_2_O →  2 CaHPO_4_ + 4H_2_O

2 CaHPO_4_ + 4H_2_O + 0.7CaO + 0.3ZnO

→  2 CaHPO_4_ + 0.7Ca(OH)_2_ + 3H_2_O + 0.3ZnO

→  Ca_2.7_(HPO_4_)_0.6_(PO_4_)_1.4_·4.4H_2_O + 0.3ZnO

### 4.2. Changes of Molecular-Interaction of the Ground S-CP Products during Wet and Dry Grinding Conditions

According to the previous section, wherein the mechanochemical synthesis of ZnAP is analyzed using XRD, S-CP is transformed into Ca-deficient HAP via an amorphous solid owing to the substantial amount of water in the wet experiment. In the dry experiment, S-CP is transformed into an amorphous solid; however, HAP is not formed owing to the limited amount of water in the solid. Therefore, the ZnAP transformation pathway by wet grinding is significantly different from that by dry grinding. To comprehensively identify differences in the synthetic pathways between wet and dry grinding at the molecular level, both grinding processes are observed in a non-destructive manner using NIRS (Figure 5), and changes in the molecular interaction mechanism are evaluated. The spectra of S-CP are altered owing to its transformation into the HAP and amorphous solid during wet and dry grinding (Figure 5b) by mechanochemical effects. The NIR spectra of each chemical component as the raw powder material (Figure 5a) are assigned based on the NIR spectral database [64] as follows: the peak at approximately 7092 cm^−1^ is attributed to the first overtone (OT) of the stretching vibration (ST) of the OH group. The peak at approximately 5200 cm^−1^ is attributed to the combination tone (CT) between ST and the deformation vibration (DF) of the OH group in crystal water. The peak at approximately 4000 cm^−1^ is attributed to the ST of the OH group in water.

The second-derivative spectra of the ground products (Figure 6) distinctly indicate changes in molecular behavior in the S-CP crystal owing to grinding. Therefore, the molecular interaction in the S-CP solid is analyzed from the change in the second-derivative SP, which can be explained by the link with the OH group during grinding. The absorption peak at 7085 cm^−1^ may be ascribed to first OT of the ST of the OH group because of the adsorbed free water on the surface of CaO [64]. The peaks at 5180 and 5110 cm^−1^ are ascribed to the CT of ST and DF of the OH group in the crystal water of CHPD, and the peak at 3876 cm^−1^ is ascribed to the second OT of the DF of the OH group in water.

In wet grinding, the peak related to CaO at 7085 cm^−1^ disappears after grinding for more than 30 min (Figure 7 and Figure 8) because CaO is transformed into Ca(OH)_2_ under wet conditions. The peak intensity at 5180 cm^−1^ of the crystal water of CHPD increases with an increase in the amount of amorphous solid or Ca-deficient HAP; additionally, the peak intensity at 3876 cm^−1^ related to the OH group is increased. This increase possibly indicates that free water in the environment under wet grinding conditions stimulates the formation of Ca(OH)_2_ and Zn(OH)_2_, and the amorphous solid formation from CHPD, accompanied by dehydration, is accelerated by mechanochemical energy. Subsequently, Zn^2+^ and Ca^2+^ react with the unstable amorphous solid, thereby transforming the solid into HAP under wet conditions.

In contrast, in dry grinding, the peak intensities at 5180 and 5110 cm^−1^ of the crystal water of CHPD gradually increase owing to amorphous solid formation with increasing grinding time (Figure 7). The peak at 7085 cm^−1^ also slowly increases with increasing grinding time because CaO and CHPD are transformed into amorphous solids under solid-state conditions. As amorphization progresses, the peak intensities at 5180 and 5110 cm^−1^ in the wet grinding process increase more slowly than those in dry grinding; however, the peak intensity at 7085 cm^−1^ in dry grinding increases more slowly than that in wet grinding. The change in the peak intensity at 3876 cm^−1^ owing to the progress of amorphization during dry grinding is approximately identical to that during wet grinding (Figure 8).

In a previous study [65], the polymorphic transformation of carbamazepine was performed in a centrifugal ball mill device controlled at 17 or 90% relative humidity (RH) during milling and analyzed based on kinetic models. In grinding the stable anhydrous polymorphs at 17 or 90% RH, an equilibrium constant crystal content of approximately 80% is attained, indicating that the grinding of the anhydrous form represents an equilibrium reaction model between the crystalline and amorphous solid forms. In contrast, for unstable dihydrate crystals under environmental conditions at 17 and 90%RH, the dihydrate form is transformed into an amorphous solid by grinding and then gradually transformed into a stable anhydrous form. This result indicates that the grinding of the dihydrate follows a consecutive reaction model to the stable form via the amorphous phase. Considering the current study on ZnAP grinding with reference to that on carbamazepine polymorphic transformation [65], the effect of grinding conditions on the crystalline pathway of the S-CP can be concluded to be as follows: All the results of molecular interactions associated with the OH groups of the ground ZnAP during wet grinding indicate that the S-CP is transformed into the ZnAP composite via the amorphous solid, because S-CP and the amorphous solid are unstable solids under wet conditions. Therefore, the mechanism may be followed a consecutive reaction model.

In contrast, S-CP and the amorphous solid are both stable solids under dry conditions; therefore, the dry grinding of S-CP may be followed an equilibrium reaction model between the crystal and amorphous solid.

## 5. Conclusions

To efficiently synthesize biomaterials with DDS functioning, ZnAP was prepared through mechanochemical treatment in wet and dry environments, and its reaction mechanisms were kinetically investigated from crystallographic and spectroscopic perspectives. S-CP consisting of CHPD, CaO, and ZnO was transformed into ZnAP with sustained Zn release by wet grinding for more than 1 h in a centrifugal ball mill; however, the complex was not formed by dry grinding. The reaction rate of the amorphization of S-CP was evaluated by measuring the XRD%. The change in the intermolecular structure of the ground S-CP products during grinding was estimated by measuring the change in second-derivative NIRS. The XRD and NIRS results suggest that the S-CP is mainly transformed into stable ZnAP via an unstable amorphous solid by wet grinding, and the reaction follows a consecutive reaction model. In dry grinding, the raw material CHPD is transformed into a stable amorphous solid by grinding, but the reaction tends to follow the equilibrium reaction model between CHPD and the amorphous solid; however, a substantial fraction of ZnO is not transformed and remains crystalline. The changes in the crystal structure and intermolecular structure of the ground products in the ZnAP mechanochemical synthesis are kinetically analyzed using XRD and NIRS. Additionally, the effect of wet conditions on the physicochemical mechanism of the ground ZnAP products during grinding is elucidated.

## Figures and Tables

**Figure 1 pharmaceutics-14-02105-f001:**
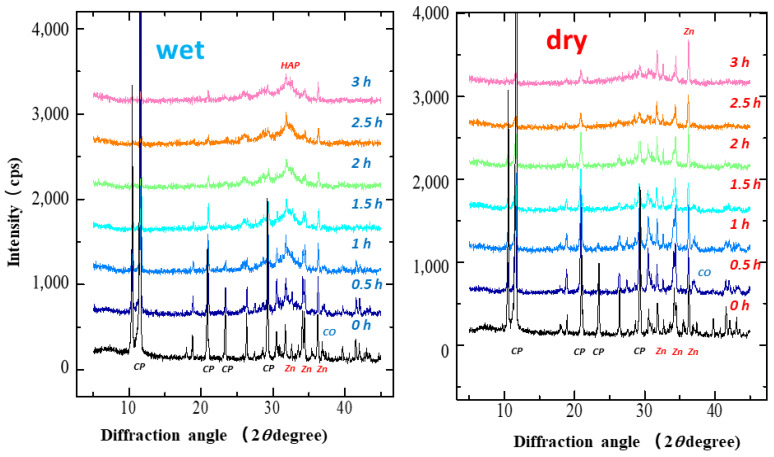
X-ray diffraction (XRD) patterns of S-CP containing CHPD (CP), CaO (CO), and ZnO (Zn) (20:7:3) during wet or dry grinding conditions. Left: wet grinding; right: dry grinding. Left is wet grinding; right is dry grinding. The powder X-ray diffraction pattern of CHPD was identified as “Brushite” (CaHPO_4_·2H_2_O) PDF#00-009-0077 (RDB) by the Powder Diffraction File^TM^ system in International Centre for Diffraction DATA, https://www.icdd.com/pdf-4-web/ (accessed 8 May 2022).

**Figure 2 pharmaceutics-14-02105-f002:**
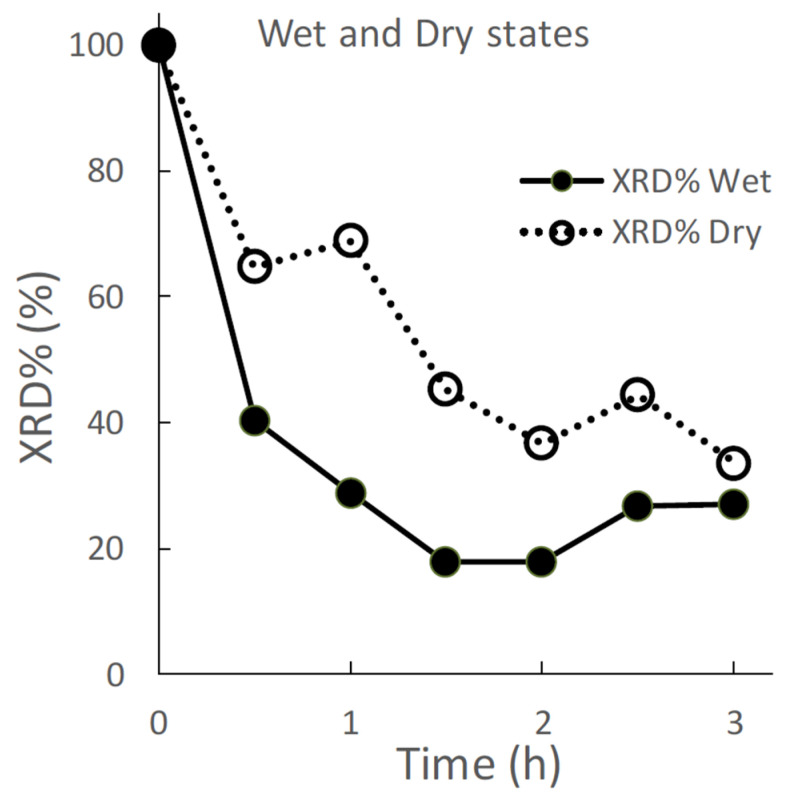
Effect of wet and dry grinding on the XRD %area (XRD%) of the ground S-CP products.

**Figure 3 pharmaceutics-14-02105-f003:**
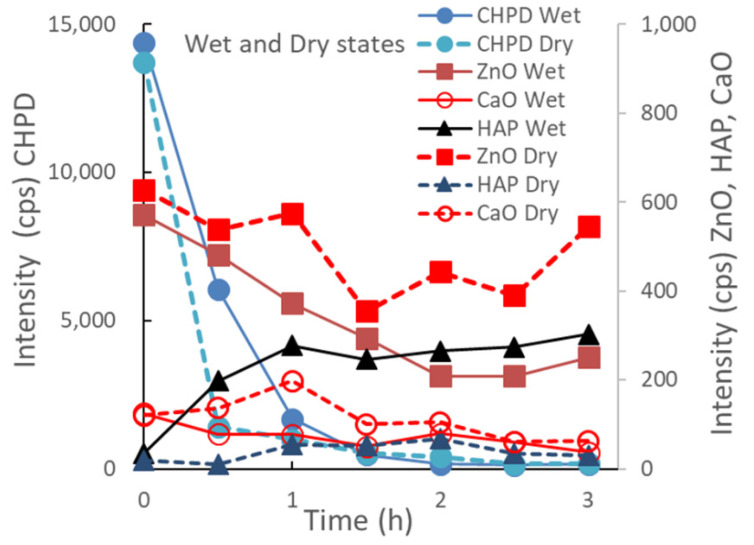
Changes in XRD peak height intensities of individual raw materials in the S-CP during wet or dry grinding.

**Figure 4 pharmaceutics-14-02105-f004:**
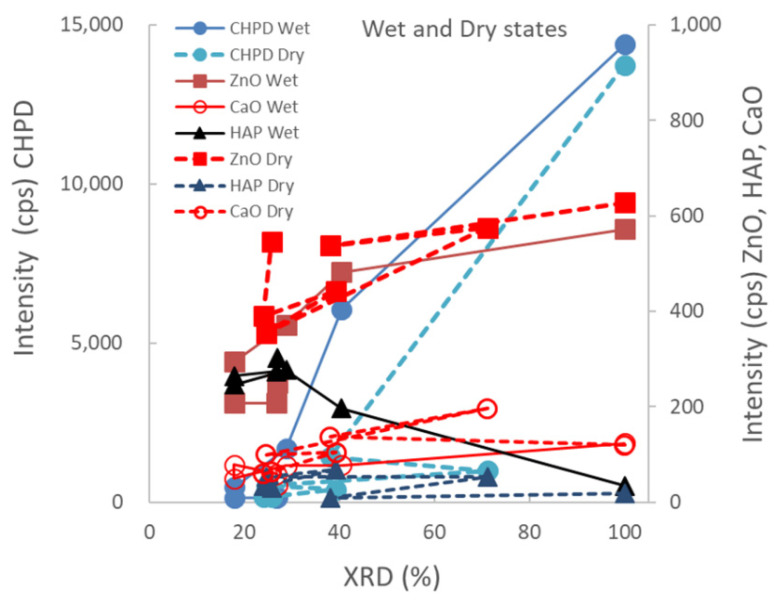
Relationship between XRD peak height intensities of individual raw materials in the ground S-CP products obtained by wet or dry grinding and their XRD%.

**Figure 5 pharmaceutics-14-02105-f005:**
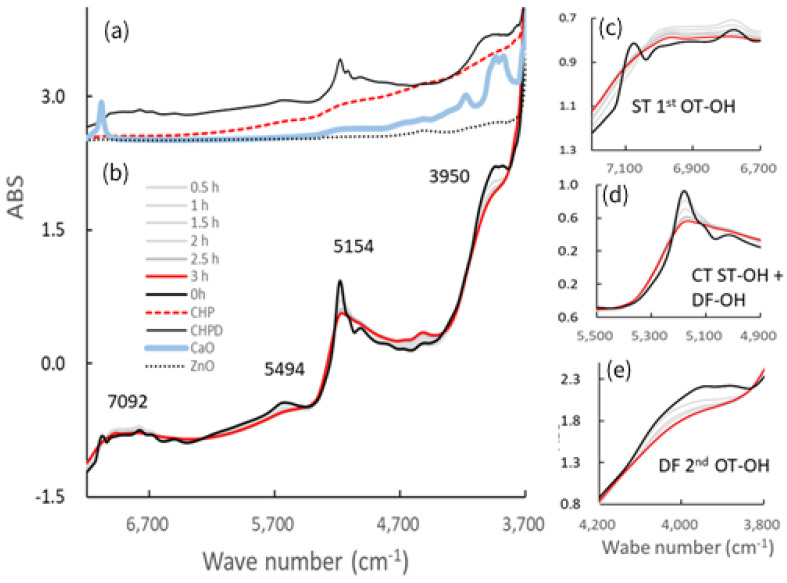
Near-infrared (NIR) spectra of the ground S-CP products obtained at various grinding times and their pure starting materials. (**a**) NIR spectra of each original starting material; (**b**) change in the spectra of the S-CP consisting of CHPD, CaO, and ZnO (20:7:3) during wet grinding; NIR spectra at wavelengths of approximately (**c**) 7000, (**d**) 5000, and (**e**) 4000 cm^−1^. OT, overtone; DF, deformation vibration; ST, stretching vibration; CT, combination tone.

**Figure 6 pharmaceutics-14-02105-f006:**
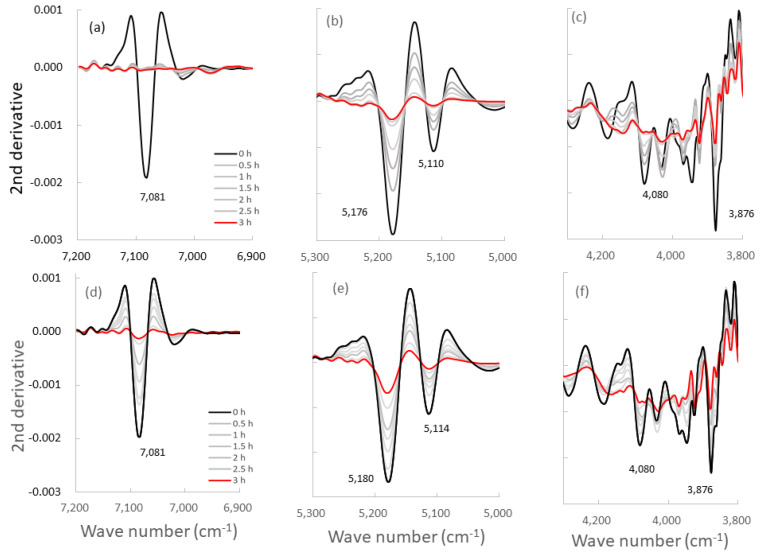
Second-derivative NIR spectra of the ground S-CP products obtained at various grinding times. (**a**–**c**): second-derivative spectra of the products obtained by wet grinding; (**d**–**f**): spectra of the products obtained by dry grinding; The spectra are at wavelengths of approximately (**a**,**d**) 7000, (**b**,**e**) 5000, and (**c**,**f**) 4000 cm^−1^.

**Figure 7 pharmaceutics-14-02105-f007:**
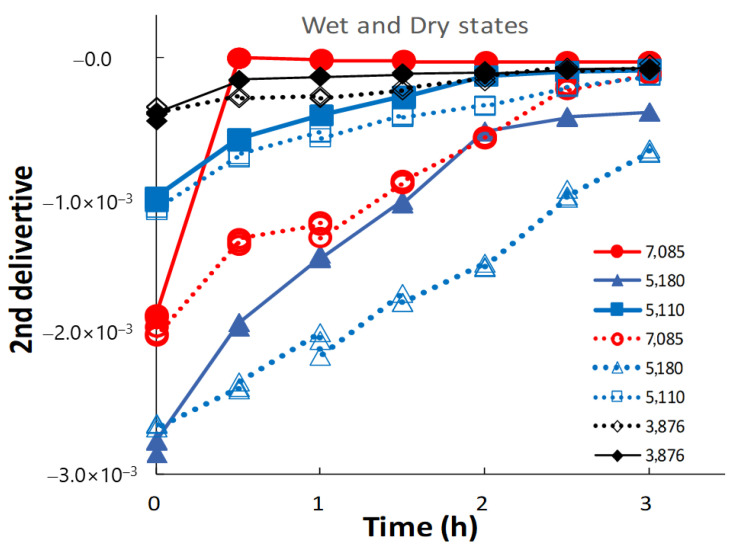
Changes in NIRS peak intensities at approximately 7000, 5000, and 4000 cm^−1^ in the second-derivative spectra of the ground S-CP products during wet and dry grinding. The solid and dotted lines indicate the changes in the intensity owing to wet and dry grinding, respectively.

**Figure 8 pharmaceutics-14-02105-f008:**
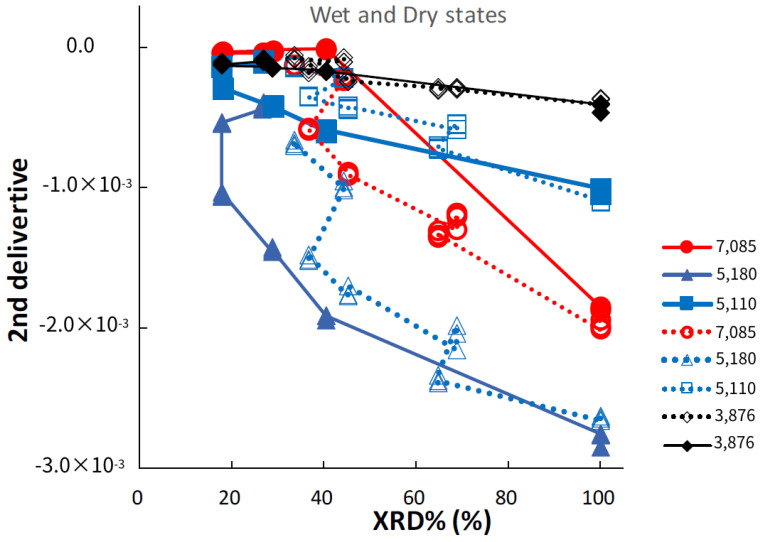
Relationships between NIRS peak intensities at specific bands in second-derivative spectra of the ground S-CP products during wet or dry grinding and their XRD%. The solid and dotted lines indicate the changes in the intensity owing to wet and dry grinding, respectively.

## Data Availability

Not applicable.

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
