# Peer review of "Analytical Evaluation of Wet and Dry Mechanochemical Syntheses of Calcium-Deficient Hydroxyapatite Containing Zinc Using X-ray Diffractometry and Near-Infrared Spectroscopy"

_pharmaceutics, 2022, doi:10.3390/pharmaceutics14102105_

Round 1
Reviewer 1 Report
The mechanochemical activation it is an interesting research field, also for hydroxyapatite based materials. To improve this paper, the authors should complete this work with some important information:
1. in generally the authors should refresh the bibliography fris of all with paper from the last 5 years (not older!)
2. line 78: in the introduction the authors mentiones the term of "green chmistry", regarding the mechanochemical activation. They should remark, that mechanochemical activation is a process with an important energy consumption!
3. line 87: why is important, which are the practical applications of the ZnAp containing "injectable suspension" ?
4. to elucidate the mechanism of the reactions it is very important to have the chemical characterizations of the samples. Only XRD and IR are not sufficient!
Author Response
Comments and Suggestions for Authors by Reviewer 1.
The mechanochemical activation it is an interesting research field, also for hydroxyapatite based materials. To improve this paper, the authors should complete this work with some important information:
- in generally the authors should refresh the bibliography fris of all with paper from the last 5 years (not older!)
èMost of the old references (25) were replaced with current references as shown in bellows:
[1] Ghiasi, B.; Sefidbakht, Y.; Mozaffari-Jovin, S.; Gharehcheloo, B.; Mehrarya, M.; Khodadadi, A.; Rezaei, M.; Siadat S. O. R.; Uskoković, V. Hydroxyapatite as a biomaterial–a gift that keeps on giving. Drug Devel. Ind. Pharm., 2020, 46(7), 1035-1062.
[5] Yang, D.; Wan, Y. Molecular determinants for the polarization of macrophage and osteoclast. In Seminars in Immunopathology, 2019, 41(5), 551-563. Springer Berlin Heidelberg.
[6] Otsuka, M. Intelligent Drug Delivery System for Artificial Bone Cement Based on Hydroxyapatite-Related Organic/Inorganic Composite Materials. In Innovative Bioceramics in Translational Medicine I, 2022, 231-253. Springer, Singapore.
[7] Otsuka, Y. Synthesis of Hydroxyapatite: Crystal Growth Mechanism and Its Relevance in Drug Delivery Applications. In Innovative Bioceramics in Translational Medicine I, 2022, 213-229. Springer, Singapore.
[8] Mabroum, H.; Noukrati, H.; Lefeuvre, B.; Oudadesse, H.; Barroug, A. Physicochemical, setting, rheological, and mechanical properties of a novel bio-composite based on apatite cement, bioactive glass, and alginate hydrogel. Ceramics International, 2021, 47(17), 23973-23983.
[9] Mabroum, H.; Noukrati, H.; Oudadesse, H.; Barroug, A. The effect of bioactive glass particle size and liquid phase on the physical-chemical and mechanical properties of carbonated apatite cement. Ceramics International, 2022, 48(19), 28207-28220.
[10] Otsuka, Y.; Ito, A.; Takeuchi, M.; Tanaka, H. Effect of amino acid on calcium phosphate phase transformation: Attenuated total reflectance-infrared spectroscopy and chemometrics. Colloid and Polymer Science, 2019, 297(1), 155-163.
[11] Meert, M. Mesoporous silica as a possible drug delivery system for tricalcium silicate cements. Master of science dissertation in the biomedical sciences). Ghent University, Belgium. 2018.
[12] Xu, C.; Wen, Y.; Zhou, Y.; Zhu, Y.; Dou, Y.; Huan, Z.; Chang, J. In vitro self‐setting properties, bioactivity, and antibacterial ability of a silicate‐based premixed bone cement. Int. J. Applied Ceramic Tech., 2018, 15(2), 460-471.
[13] Otsuka, M., Basic Sciences to support Bio-Integration: Development of hybrid medical systems supported by Medical, Dental, Pharmaceutical and Industrial Sciences., J. Bio-Integration 2019, 9 (1) 1-8,
[15] Chung, H.; Bird, A. J. Zinc Signals in Biology. In Zinc Signaling 2019, 389-410. Springer, Singapore.
[32] Chamniansawat, S.; Kampuang, N.; Suksridechacin, N.; Thongon, N. Ultrastructural intestinal mucosa change after prolonged inhibition of gastric acid secretion by omeprazole in male rats. Anatom. Sci. Int., 2021, 96(1), 142-156.
[33] Mousa, S. O.; Abd Alsamia, E. M.; Moness, H. M.; Mohamed, O. G. The effect of zinc deficiency and iron overload on endocrine and exocrine pancreatic function in children with transfusion-dependent thalassemia: a cross-sectional study. BMC pediatrics, 2021, 21(1), 1-9.
[34] Maares, M.; Keil, C.; Straubing, S.; Robbe-Masselot, C.; Haase, H. Zinc deficiency disturbs mucin expression, O-glycosylation and secretion by intestinal goblet cells. Int. J. Molecular Sci., 2020, 21(17), 6149.
[36] Harikrishnan, P.; Sivasamy, A. Preparation, characterization of Electrospun Polycaprolactone-nano Zinc oxide composite scaffolds for Osteogenic applications. Nano-Structures & Nano-Objects, 2020, 23, 100518.
[37] Nakano, M.; Nakamura, Y.; Miyazaki, A.; Takahashi, J. Zinc Pharmacotherapy for Elderly Osteoporotic Patients with Zinc Deficiency in a Clinical Setting. Nutrients, 2021, 13(6), 1814.
[38] Huang, T.; Yan, G.; Guan, M. Zinc homeostasis in bone: zinc transporters and bone diseases. Inter. J. Molecular Sci., 2020, 21(4), 1236.
[45] Japan Chemical Industry Association, Efforts against global warming in the chemical industry, accessed September 2 2022. chrome-extension://efaidnbmnnnibpcajpcglclefindmkaj/https://www.meti.go.jp/shingikai/sankoshin/sangyo_gijutsu/chikyu_kankyo/kagaku_wg/pdf/2019_01_04_01.pdf
[46] Millennium declaration by the United Nations 2000, accessed September 2 2022. https://www.mofa.go.jp/mofaj/kaidan/kiroku/s_mori/arc_00/m_summit/sengen.html
[47] Borges, R.; Giroto, A. S.; Klaic, R.; Wypych, F.; Ribeiro, C. Mechanochemical synthesis of eco-friendly fertilizer from eggshell (calcite) and KH2PO4. Ad. Powder Tech., 2021, 32(11), 4070-4077.
[48] Geethakarthi, A. Synthesis of Hydroxyapatite Nanoparticle from Papermill Sludge. Textile Wastewater Treatment, 2022, 311-328.
[49] Swain, S.; Bhaskar, R.; Gupta, M. K.; Sharma, S.; Dasgupta, S.; Kumar, A.; Kumar, P. Mechanical, electrical, and biological properties of mechanochemically processed hydroxyapatite ceramics. Nanomaterials, 2021, 11(9), 2216.
[50] Chesley, M.; Kennard, R.; Roozbahani, S.; Kim, S. M.; Kukk, K.; Mason, M. One-step hydrothermal synthesis with in situ milling of biologically relevant hydroxyapatite. Mat. Sci. Eng. 2020, C, 113, 110962.
[51] Bulina, N. V.; Chaikina, M. V.; Prosanov, I. Y. Mechanochemical synthesis of Sr-substituted hydroxyapatite. Inorganic Materials, 2018, 54(8), 820-825.
[56] Saeys, W.; Do Trong, N. N.; Van Beers, R.; Nicolaï, B. M. Multivariate calibration of spectroscopic sensors for postharvest quality evaluation: A review. Postharvest Biology and Technology, 2019, 58, 110981.
[64] Ozaki, Y.; Huck, C.; Tsuchikawa, S.; Engelsen, S. B. (Eds.). Near-infrared spectroscopy: theory, spectral analysis, instrumentation, and applications (pp. 978-9811586477). 2021, Berlin/Heidelberg, Germany: Springer.
- line 78: in the introduction the authors mentions the term of "green chemistry", regarding the mechanochemical activation. They should remark, that mechanochemical activation is a process with an important energy consumption!
èThe sentences related to “Green chemistry” are added in the text, page 2, line 77-87, as “By the way, wet chemical reactions which consume a large amount of energy, emit a large amount of CO2, and emit a large amount of highly toxic organic solvents, waste liquids, and exhaust gas are became a hindrance factor for the world environmental [45]. The United Nations has recommended industrial use of green chemistry to reduce negative impacts on the future global environment as a Sustainable Development Goal [46]. Chemical reactions based on mechanochemistry could be recommended as ideal green chemistries without high energy consumption per unit weight or producing waste solvents. The solid-state reactants in mechanochemical synthesis [47-50] are ground and absorbing the minimum energy required to initiate the reaction in a ball mill and efficiently proceeding with the reaction to the product without any toxic waste solvents and gases.”.
è The related 2 references are added in the text as
[45] Japan Chemical Industry Association, Efforts against global warming in the chemical industry, accessed September 2 2022. chrome-extension://efaidnbmnnnibpcajpcglclefindmkaj/https://www.meti.go.jp/shingikai/sankoshin/sangyo_gijutsu/chikyu_kankyo/kagaku_wg/pdf/2019_01_04_01.pdf
[46] Millennium declaration by the United Nations 2000, accessed September 2 2022. https://www.mofa.go.jp/mofaj/kaidan/kiroku/s_mori/arc_00/m_summit/sengen.html
- line 87: why is important, which are the practical applications of the ZnAp containing "injectable suspension" ?
èThe sentences related to importance of "injectable suspension" are added in the text, page 2, line 93-101, as “An injectable suspension formulation using calcium phosphate nanoparticles containing Zn can be applied to any parts of the body. At the injection site, the nanoparticles with bioaffinity are phagocytosed by cells such as macrophages, and calcium phosphate dissolved and release Zn [54], so, sustained Zn release rate could be freely controlled. These results indicated that the injectable suspension formulation of calcium phosphate containing Zn may be useful practical dosage form for the slow release of Zn.”.
- to elucidate the mechanism of the reactions it is very important to have the chemical characterizations of the samples. Only XRD and IR are not sufficient!
èIn the previous report [54], the crystalline properties of Zn contained complexes and its formation were qualitatively characterized by various analytical methods including powder X-ray analysis, scanning electron microscopy measurement with energy-dispersive X-ray spectroscopy, the specific surface area measurement with Brunauer–Emmett–Teller nitrogen gas adsorption, and In vitro Zn release test. In this study, therefore, the changes in crystallinity were analyzed quantitatively using XRD and NIR methods, and their formation processes were kinetically reported.
Reviewer 2 Report
The manuscript describes the influence of the wet or dry grinding environment on the synthesis mechanism of the calcium phosphate complex containing Zn from the combined raw materials. The crystal structure and intermolecular structure were kinetically analyzed using powder X-ray diffraction (XRD) and near-infrared spectroscopy (NIRS). The manuscript is well written and the results are interesting!
Here are my comments:
- The authors should emphasize the novelty of the subject in the Introduction section.
- JCPDS card should be added in Fig. 1 so that the XRD results can be better explained.
- Figures should be inserted into the manuscript and the graphical abstract
Author Response
Comments and Suggestions for Authors by Reviewer 2.
The manuscript describes the influence of the wet or dry grinding environment on the synthesis mechanism of the calcium phosphate complex containing Zn from the combined raw materials. The crystal structure and intermolecular structure were kinetically analyzed using powder X-ray diffraction (XRD) and near-infrared spectroscopy (NIRS). The manuscript is well written and the results are interesting!
Here are my comments:
- The authors should emphasize the novelty of the subject in the Introduction section.
èThe novelty of the subject was added in the Introduction on page 2, line 93-108, as “An injectable suspension formulation using calcium phosphate nanoparticles contain-ing Zn can be applied to any parts of the body. At the injection site, the nanoparticles with bioaffinity are phagocytosed by cells such as macrophages, and the particles dis-solved and release Zn [54], so, sustained Zn release rate could be freely controlled. The result indicated that the injectable suspension formulation of calcium phosphate na-noparticles containing Zn may be useful practical dosage form for the slow release of Zn. However, in mechanochemical synthesis of Zn-releasing nanoparticles, the rela-tionship between molecular chemical mechanisms and environmental factors for wet and dry milling has not been elucidated yet. In this study, therefore, the effects of the wet or dry grinding environment on the synthetic mechanism of the calcium phos-phate complex containing Zn from the mixed powder of the raw materials and their molecular interactions were investigated using kinetic methods based on powder X-ray diffraction (XRD) and near-infrared spectroscopy (NIRS).”.
- JCPDS card should be added in Fig. 1 so that the XRD results can be better explained.
èJCPDS data was added in the text on page 4 as“The powder X-ray diffraction pattern of CHPD was identified as "Brushite" PDF#00-009-0077 (RDB)) by the Powder Diffraction FileTM system in International Centre for Diffraction DATA, https://www.icdd.com/pdf-4-web/.”, and Figure 1 caption was corrected.
- Figures should be inserted into the manuscript and the graphical abstract
èFormat of the manuscript was corrected and all of Figures were inserted in the text, following your suggestion.
Round 2
Reviewer 1 Report
Thank you for the corrections. Please put also some EDS-results also in the paper related to the samples!
Author Response
uthor's Reply to the Review Report (Reviewer 1)
Thank you for the corrections. Please put also some EDS-results also in the paper related to the samples!
Thank you very much for your good suggestion. We partially agree with your opinion that adding new EDS data in the text to improve quality of the research manuscript. However, currently, I just moved new position, Shizuoka University, and it is impossible to add new data in short period, such as within 5 days. Additionally, as I mentioned in the text, the mechanochemical conversion kinetics of ZnAP by XRD and NIR is the quantitative study as the second paper, since the qualitative study on the mechanochemical conversion was reported as the first paper. In order to solve these problems, we would like to ask for the understanding of the readers and reviewers by describing and modifying the relationship of mechanochemical treatment of ZnAP between the previous report and this research report as follows.
The related sentences were corrected on page 3, line 91-103, as “Crystalline properties of ZnAP were sufficiently characterized by qualitative methods, such as powder X-ray analysis, scanning electron microscopy with energy-dispersive X-ray spectroscopy, the nitrogen gas adsorption and in-vitro Zn release test, and the results clearly indicated that the formation of Ca-deficient HAP containing Zn during grinding [54]. Biomedical activities of the ZnAP were optimal ranges for bone regeneration medicine by evaluating various pharmaceutical properties. An obtained injectable suspension formulation using ZnAP can be applied to any parts of the body. At the injection site, the nanoparticles with bioaffinity are phagocytosed by cells such as macrophages, and the particles dissolved and release Zn [54], so, sustained Zn release rate could be freely controlled. The result indicated that the injectable ZnAP suspension formulation may be useful practical dosage form for the slow release of Zn. However, the detailed molecular formation mechanism in the wet and dry mechanochemical synthesis of therapeutically useful ZnAP has not been clarified yet.”.